# Evaluation of Novel Ornamental Cladding Resistance, Comprised of GFRP Waste and Polyester Binder, within an Acid Environment

**DOI:** 10.3390/polym13030448

**Published:** 2021-01-30

**Authors:** Emilia Sabău, Paul Bere, Mărioara Moldovan, Ioan Petean, Cristina Miron-Borzan

**Affiliations:** 1Department of Manufacturing Engineering, Machine Building Faculty, Technical University of Cluj-Napoca, 103–105 Muncii Bd., 400641 Cluj-Napoca, Romania; Emilia.Sabau@tcm.utcluj.ro (E.S.); Paul.Bere@tcm.utcluj.ro (P.B.); 2Raluca Ripan Chemistry Research Institute, Babeş-Bolyai University, 30 Fântânele Str., 400294 Cluj-Napoca, Romania; mmarioara2004@yahoo.com; 3Department of Chemical Engineering, Faculty of Chemistry and Chemical Engineering, Babeş-Bolyai University, 11 Arany Janos Str., 400084 Cluj-Napoca, Romania; ipetean@chem.ubbcluj.ro

**Keywords:** composite materials, waste glass fibers, polyester matrix, chemical attack, surface analysis, surface roughness, X-ray diffraction, mineral analysis

## Abstract

The paper presents the manufacturing technology for a material obtained from glass fiber waste, quartz sand, and polyester binder, used for ornamental building plates. The composite has a cover surface that ensures protection of the material from environment attacks and a structural material that can be subjected to chemical degradation. The mechanical properties of the obtained material were experimentally investigated through compressive mechanical tests. To observe the material’s behavior in contact with external agents (rain or acid rain, due to environmental pollution), analyses were performed in laboratory conditions. An investigation on the effects of chemical attack substances was conducted. SEM and macroscopic analyses were performed, and the surface roughness was determined for each sample area. The obtained results were statistically analyzed and showed that there is no significant difference between the surface roughness for treated and untreated samples. Furthermore, the surfaces were analyzed by X-ray diffraction and mineralogical optical microscopy in polarized light with crossed nicols. It was observed that rainwater does not affect the plate structure even if the plates are used in high-pollution environments. The material is suitable for exterior building walls from the point of view of chemical attack and resistance.

## 1. Introduction

For centuries, stone has been considered as being the perfect material for ornamental plates building thanks to its durability, aesthetics, and availability. However, it was found that the most frequent and dangerous degradation phenomena are correlated to the presence of water [1,2].

In the manufacturing process of ornamental synthetic plates used in different applications, sand mixtures with different binders such as whitewash or cement and plaster [3,4] are known to be used. The dry or wet material is pressed into a mold and extracted after polymerization. There are many studies on the manufacturing of the reinforced mortars used in construction and used for designing decorative panels which meet aesthetic and quality requirements [5,6,7,8,9,10]. These mortars include in the mixture, besides sand, cement, whitewash, and gypsum, various reinforcement materials such as glass fibers, hemp fibers, wood fibers, sand, etc.

An alternative solution can be the use of composite materials (CMs) that offer retrofitting and designing possibilities for building walls. CMs made of a reinforcement material (glass fibers, carbon, Kevlar, etc.) and a matrix (polyester and epoxy resin, etc.) continuously develop, as they are used in products designed for daily life use as well. Composites may also contain some auxiliary materials such as catalysts, accelerators, couplings agents, pigments, fillers, etc., to provide improved properties or modification of the desired properties [11,12,13,14,15]. Experimental results led to the determination of the durability, mechanical behavior, protective effect, and thermal compatibility of mortars. The weathering resistance of polymer mortars was also presented in [16,17,18].

The corrosion resistance and durability of glass fiber-reinforced polymer material were analyzed in [19], and it was found that they depend on the type of glass fiber used. The combined effect of moisture, alkali, and temperature parameters and also the rate magnitude of damage of glass fiber-reinforced polymer (GFRP) composites were studied by [20].

Fiber-reinforced polymers (FRPs) have been found to be particularly attractive for their durability considerations and their applications, mainly for the reinforcement of existing structures [21,22,23]. Polini et al. [24] analyze the use of glass/epoxy composite laminates as external reinforcements in comparison with natural stones, highlighting the advantages of composites (better mechanical resistance and lower weight). The mechanical properties of natural stones in comparison with different types of GFRP structures from glass/epoxy laminates and a honeycomb or foam core are also presented in [25,26,27]. Pickering [28] presented recycling techniques (grinding technologies) in order to obtain different size ranges of fillers or partial reinforcement for composite materials. Different authors have also presented recycling materials and techniques to obtain new construction products [29,30,31,32,33].

It is important to know the behavior of composites in different corrosive environments; therefore, in the scientific literature, there are studies regarding this subject. In [34], the authors presented a study on natural stone reconstruction to see the behavior in marine environments. A study on composite structures exposed to a range of corrosive environments which cause degradation in terms of material properties was conducted by other authors [35,36]. The effect of different solutions (alkaline and acid) on glass/epoxy composites’ mechanical behavior was analyzed. A comparative study on the influence of chemical degradation effects on mechanical tests of epoxy polymer mortars was conducted by Reis [37], using eight degradation agents found in corrosive processes from industrial environments. Gorninski et al. [38] have performed an evaluation of the chemical resistance of different mixtures of polymeric mortars by exposing them to corrosive agents that can be found in industrial environments. Gutberlet et al. [39] investigated the influence of acid on wetted cement. In a study conducted by other researchers [40,41], it was found that glass fiber from reinforced epoxy resin materials can be separated under the influence of amine in a nitric acid and hydrogen peroxide solution. Some researchers investigated the corrosion rate of cement mortars in nitric acid by measuring the corrosion depth [42] and the corrosion in nitric and acetic acids with different concentrations for hardened cement paste, taking into consideration the time of action and the depth of corrosion [43].

In the case of ornamental plates, the best orientation of the stone is essential to avoid greater damage, and this leads to a precise assembly. In order to reduce the assembly time, a new plate with higher dimensions should contain several assembled stones made of CM with glass fiber waste. At the same time, the calcite is very much used in the construction domain for facades made from calcareous stone and marble. It is known that calcite is chemically eroded by nitration (HNO_3_H_2_SO_4_) and acetic acid (CH_3_COOH). It is very important to develop calcite resistance to COOH.

Taking all of this into consideration, as well as the previous research in the field and the fact that there are no other studies on this material, the purpose of this study was to obtain a new material for building plates and to test their resistance in acid environment conditions. The material was obtained from recycling glass fiber waste mixed with quartz sand and polyester binder in different weight fraction ratios. We investigated the most unfavorable hypothesis-the infiltration of rainwater at the edges of the structure due to exposure to the external environment. In order to determine its behavior in environmental conditions, the material surface was attacked by a series of chemical solvents. It is considered that the solvents used can simulate acid rain or external corrosive factors. The roughness values obtained were evaluated for the initial structure and that after acid attacks: rainwater, acetic acid and nitric acid. Therefore, a scanning electron microscopy analysis was performed for each area and for the interface between the ornamental surface and the resistance structure.

The novelties of this study are the evaluation of this new material obtained from GFRP waste and research regarding the behavior of the material under chemical attack.

In order to evaluate the possible modifications under the chemical attack, a morphological analysis using a scanning electron microscope (SEM) and atomic force microscopy (AFM) was performed and is presented in this study.

The roughness results were statistically analyzed for the interior and exterior parts of the samples. X-ray diffraction (XRD) analysis and mineralogical optical microscopy in polarized light with crossed nicols (CPOM) analysis were also used for a deep investigation.

## 2. Materials and Methods

### 2.1. Materials

The material for the building plates was obtained by grinding glass fiber waste, calcium carbonate (CaCO_3_), and sand (quartz) in polyester matrix. The waste was obtained from companies which produce composite materials. The glass fiber wastes were ground using a GSL 300 Granulator from ZERMA GmbH, Zuzenhausen, Germany. The granulator is used for recycling different plastic or CM waste. The material grinding was performed based on specially profiled rotor knives which have a slow speed (150 rpm). The rotor diameter of the granulator was 300 mm and rotor width 400 mm. It used a screen size of up to 6 mm.

After the grinding procedure of the GFRP, the resulting material contained small particles of polyester and agglomerate glass fiber monofilaments in random positions (glass wool).

This proposed material contains two polymeric CMs. The first is surface layer and contains 63% polyester matrix and 37% CaCO_3_ in weight fraction ratio (wf). The CaCO_3_ was in the form of dust, up to 2 µm in particle size. The polyester resin used was Polylite 32166-16 type and 1.25% Norpol peroxide 10. The polyester matrix was delivered from Reichhold CZ s.r.o., Ústí nad Labem, Czech Republic. The second is structural composite, which ensures the resistance of the first layer. The structural CM is composed of 38% wf polyester matrix, 31% wf sand, up to 0.3 mm sort range, and 31% wf grinded glass fiber waste.

### 2.2. Manufacturing Methodology of New Composite

A silicon mold was used in order to obtain ornamental building plates. The surface layer material copied the mold and took the appearance of a synthetic stone. The polyester matrix and CaCO_3_ were mixed for 5 min. The composition was cast on the silicon mold and maintained until the gel point time. The second mixtures of structural CM, polyester matrix, quartz sand, and grinded glass fiber waste were mixed for 10 min in order to achieve homogenization of components. The materials were cast over the first layer and maintained in the mold until polymerization [44,45]. The mixing of the materials was performed at 20 °C in the laboratory condition. Supplementary curing procedures were applied at 80 °C for 4 h in the oven.

The reinforced materials and the manufacturing procedures have a significant influence on the composite structure’s quality (from the point of view of the properties). The obtained ornamental synthetic plate stone is a compact material, and the manufacturing is easy to perform.

The main scheme of the layers to obtain an ornamental synthetic plate that uses waste from GFRP is presented in Figure 1 [44]. On a support plate (1) made from GFRP is fixed the silicon mold (2). On the silicon mold is applied the release agent. This was polyvinyl alcohol, which provides a good release of the GFRP to be formed in the mold. The mold (2) is covered with the surface layer (4), consisting of a mixture formed of polyester matrix and calcium carbonate. In order to manufacture the resistance structure, a reinforcing material (5) is applied, which is composed of a mixture of sand, ground waste glass fibers, and polyester matrix in an unpolymerized state. After polymerization of the polyester matrix, a rigid synthetic material is obtained, which perfectly copies the mold shape. After mold release, an ornamental synthetic plate stone is obtained with a compact material.

An ornamental synthetic plate is presented in Figure 2. The surface layer can be colored in the initial stage by different pigments in order to obtain the natural look of stone. The average size of the thickness for the obtained plate was 15 mm. The thickness was influenced by the unregulated shape of the mold. The mold copies the shape of broken stones and forms a large plate. The average size of the thickness for the surface layer was 3 mm and varied from 1 to 6 mm, depending on the shape of the mold. For the surface layer, a porosity of 0.17% was obtained and the density was 1.757 g/cm^3^. For the structural layer, the porosity was up to 31% and the density 1.681 g/cm^3^. Very close results for porosity but for different materials and components are presented in [46]. Some researchers [47] have reduced the porosity by introducing different additives for recycled concretes.

### 2.3. Compressive Mechanical Tests

Uniaxial compressive tests were carried out according to the EN 12320-3 standard for the obtained material. Compressive tests were performed with a universal testing machine Instron 1196 (Darmstadt, Germany) equipped with a 25 kN load. Cubic specimens with 50 × 50 × 50 mm dimensions were cut from the plate using a diamond disc. The test speed was 2 mm/min. All tests were performed at room temperature.

### 2.4. Chemical Substances Influence Evaluation of the Composite Plate

In order to evaluate the behavior of the obtained ornamental synthetic plate in contact with external agents, chemical attacks were applied where three different agents were used: (i) rainwater, (ii) acetic acid, and (iii) sulfuric + nitric acid.

The chemical reactions were:
CaCO_3(s)_ + H_2_SO_4(aq)_→CaSO_4(aq)+_CO_2(g)_ + H_2_O_(l)_;
CaCO_3(s)_ + 2HNO_3(aq)_→Ca (NO_3_)_2(ag)+_ H_2_O_(l)_;
CaCO_3(s)_ + CH_3_COOH→H_2_O + CO_2_ + Ca (CH_3_COO)_2_.


The structure of the plate was analyzed before and after the acid attacks. Four different samples for each case were analyzed in five different points.

Samples from the ornamental synthetic plate were cut with a Buehler IsoMet 1000 Precision Saw machine (Esslingen, Germany) that used IsoMet Diamond Wafering Blades 152 × 0.5 mm; Arbor size: 12.7 mm.

The corrosive substances were:Case 1: Rainwater, pH = 6 (RW);Case 2: Solution of 1:1: 5 mL acetic acid + 5 mL distilled water and ammonia (18 drops) for adjustment, pH = 3.5 (AA);Case 3: Solution of 18 mL distilled water + 3 mL concentrated sulfuric acid + 2 mL concentrated nitric acid + ammonia, until pH = 2 (SNA).

The samples were stored at room temperature for 7 weeks in the corrosive substances.

### 2.5. Morphological Analyses

Scanning electron microscope and atomic force microscopy samples’ evaluation and roughness measurements were performed. SEM evaluation was performed with Inspect S model (FEI Company, Hillsboro, OR, USA), functional in high vacuum and low vacuum, with accelerating voltage between 200 V and 30 k.

In order to determine the effects of chemical attack on the composite material used, morphology surfaces were investigated. Atomic force microscopy was performed in tapping mode using a JEOL JSPM 4210 Scanning Probe Microscope, Tokyo, Japan. The samples’ surfaces were scanned with an NSC 15 cantilever produced by MikroMasch Co. from Tallinn, Estonia, having a resonant frequency of 325 kHz and a spring constant of 40 N/m. Topographic images were recorded at a scanned area of 10 × 10 µm at a scanning rate of about 1.5 Hz. The images were processed in the standard manner using the professional software Win SPM Processing 2.0 designed for Jeol AFM Microscopes. For each sample, 5 different macroscopic areas were investigated. The surface roughness average (Ra) and root mean square roughness (Rq) were measured for each of them.

### 2.6. X-ray Diffraction Analysis (XRD)

X-ray diffraction analyses were performed both for the exterior and interior areas of the samples in order to highlight the main compound and its variation after exposure to a corrosive environment. The analyses were conducted on a BRUKER D8 Advance X-ray diffractometer (Billerica, Massachusetts, United States), using CuKα (λ = 0.154 nm) radiation. XRD patterns were registered in the 20 to 85 2- θ range.

### 2.7. Mineralogical Optical Microscopy in Polarized Light with Crossed Nicols (CPOM)

The materials (exterior and interior areas) were analyzed with the help of a mineralogical Laboval 2, Karl Zeiss Jena Microscope (Jena, Germany) with a Kodak 10 MPx digital camera. The samples were polished up to 800 μm thickness in order to become transparent in polarized light.

## 3. Results and Discussion

The obtained data at compressive tests for the composite material indicate values of compression strength which are presented in Table 1. The five samples are denoted as S1, S2, S3, S4, and S5.

During the compressive mechanical tests, it could be observed that the constituents of the composite materials remained bonded through filaments.

The experimental data show that the materials have good mechanical properties. If the results are compared with a concrete material that is also used as ornamental plates (average compressive strength 58.6 MPa [48]), it can be seen that the proposed composite material has better resistance, and it can be successfully used for the purpose of ornamental plates.

SEM analyses were performed for the CM surface layer, CM structural layer, and the interface between both of the components (shown in Figure 2). The components of the CM studied are presented in the SEM investigation in Figure 3. All the components are identified and a good compatibility between the matrix and the solid components can be observed.

A remarkable connection can be observed between the monofilaments of the glass fibers and the polyester matrix covers. The polyester matrix integrates the components in the structure. The porosity of the structural material contributes to the mechanical coupling of the adhesive to bonding of the plates.

The SEM images given in Figure 4 show the surface layer before and after chemical treating of the samples with different substances. For this type of composite, the most important issue is to observe a neutral behavior of the chemical attacks on the surface area. The cover surface ensured protection to the material from environmental attacks. Chemical substances can create cracks on the surface layer of a material, and from time to time, in winter, cracks are developed and the material fails.

The material compounds are polyester resin and CaCO_3_. As it can be seen, the most affected surface is in the acetic acid (Figure 4c) and sulfuric + nitric acid (Figure 4d) cases. The CaCO_3_ particles from the surface layer dissolved in the solution of nitric acid and sulfuric acid. The created porosity does not have the capillarity to permit water infiltration. There are only individual surface pores.

To improve surface quality against chemical attacks, the CaCO_3_ fraction ratio can be decreased for the first surface layer. In this case, the Ca particles are not in direct contact with some acids and thus avoid surface degradation. The polyester polymer protects the surface of the exterior layer from environment degradation.

In Figure 5, SEM analyses of the structural CM under chemical attack are shown. The structural material can be subjected to chemical degradation in case of damage to the surface layer. The structure of the material is homogenous. Composites such as quartz particles (sand) in the polyester matrix and glass fibers can be observed. It can be observed that the constituents of the materials are well homogenized, and the polymer matrix integrates the quartz particles and the glass fibers. The structure is compact, and porosity is not presented. In all of the presented microstructures, it can be observed that all of the constituents are interconnected and integrated in the structure. In Figure 5d, the polyester matrix can be observed on the glass fiber monofilament. The polyester creates a good connection on the glass fiber surface.

The interface between the surface layer and the structural CM is investigated in Figure 6. We marked lines which indicate the border between the surface layer and the structural CM. In Figure 6c, all materials components such as glass fiber monofilaments, quartz particles, CaCO_3_, and the matrix (Figure 6d) can be observed. A good connection between the surface layer and the structural CM can be observed. The porosity indicated in Figure 6c can be a degradation source. This can lead to water infiltration and delamination of the components. The solution to eliminate material porosity is to use pressure and good compaction of the wet materials in the initial manufacturing phase.

An influence of chemical substances on the interface between these two important materials, i.e., surface layer and structural CM, could not be observed. The same conclusion can be noted in the structural CM case where components such as glass, polymer, and quartz particles are neutral from a chemical point of view. Here, the materials have porosity from the manufacturing process. This was chosen to have a good mechanical connection with cement-based adhesive for sticking on buildings. Porosity can be influenced by the delamination degradation process of materials in time. For the moment, this phenomenon cannot be observed after chemical substance exposure.

The surface topography evolution of the ornamental surface layer observed by AFM is presented in Figure 7a–d. The control group (e.g., un-etched surface) shows a microstructure typical for polished calcite. Such topography was observed for marble in the literature [49]. Exposure to rainwater does not affect the ornamental surface; Figure 7b.

Exposure to acid solutions induces several changes to the ornamental surface topography. Calcite erosion in the surface leads to the enhancement of surface depressions (e.g., pore-like), a fact observed in Figure 7c,d. The sulfuric and nitric acid solution is more erosive than acetic acid, featuring larger pores.

The topographic images obtained for the structural layer are displayed in Figure 7e–h. The control group, Figure 7e, revealed a quartz particles-based morphology, with particles having a equiaxed–polyhedral shape covered with the binder material. A significant porosity is observed between the particles. Rainwater exposure does not affect the topography of the sample, but acid exposure increases the material porosity (pore size and depth). This fact is sustained by the increased roughness values for the samples subjected to the acid treatment. The porous morphology of the structural layer makes it more sensitive to the erosion than the surface layer. The higher compactness of the surface layer leads to a higher erosion resistance than the one observed for the inside area.

A quantitative indicator for erosion is given by the surface roughness, which is mostly increased in the sample exposed to the sulfuric and nitric acid solution.

A statistical analysis was performed on Ra and Rq measured for topographic images obtained at a scanning area of 10 × 10 μm, where Ra is the arithmetic average of the absolute values of the profile heights over the evaluation length, and Rq is the root mean square average of the profile heights over the evaluation length.

Two hypotheses were considered in the statistical analysis: (H_0_) There is no significant difference in roughness between the control group and the tested group; (H_1_) There is a significant difference in roughness between the control group and the tested group. To test these hypotheses, a two-tailed independent sample t-test was conducted with a significance level of α = 0.05.

The Ra value was different in the control group for the surface layer (mean (M) = 121.960 µm, Standard Deviation (SD) = 51.109) and for the interior surface (M = 259.200 µm, SD = 100.728).

Furthermore, the Rq value was different in the control group for the surface layer (M = 154.04 µm, SD = 58.54) and for the structural layer (M = 313.600 µm, SD = 116.637).

The obtained values and descriptive statistics (mean and standard deviation) for each sample group taken into consideration are centralized in Table 2 for surface layer (the ornamental surface) and in Table 3 for structural layer (the resistance structure). A comparison is presented between the control group average and the test group averages for the surface and structural layers of the samples.

In the case of the surface layer, there was no significant difference in the values for roughness Ra of the control group (M = 121.960 µm, SD = 51.109) and the roughness of the surface treated with:-Rainwater (M = 128.260, SD = 66.643); t (4) = 0.172, *p* = 0.871;-Acetic acid solution (M = 188.200, SD =119.144); t (4) = 1.186, *p* = 0.301;-Nitric acid solution (M = 247.200, SD = 152.165); t (4) = 1.820, *p* = 0.142.

Furthermore, for the surface layer, there was no significant difference in the values for roughness Rq of the control group (M = 154.04 µm, SD = 58.54) and the roughness of the surface treated with:-Rainwater (M = 164.40, SD = 82.28); t (4) = 0.236, *p* = 0.8249;-Acetic acid solution (M = 237.20, SD = 142.59); t (4) = 1.252, *p* = 0.279;-Nitric acid solution (M = 307.00, SD = 168.64); t (4) = 2.015, *p* = 0.114.

In Table 3, a comparison is presented between the control group average and the test group averages for the structural area of the samples.

In the case of the structural layer, there was no significant difference in the values for roughness Ra of the control group (M = 259.200 µm, SD = 100.728) and the roughness of the surface treated with:-Rainwater (M = 250.180, SD = 143.445); t (4) = 1.183, *p* = 0.912;-Acetic acid solution (M = 315.600, SD = 96.984); t (4) = 0.955, *p* = 0.393;-Sulfuric + nitric acid solution (M = 333.000, SD = 117.792); t (4) = 1.138, *p* = 0.319.

For the structural layer, there was no significant difference in the values for roughness Rq of the control group (M = 313.600 µm, SD = 116.637) and the roughness of the surface treated with:-Rainwater (M = 302.660, SD = 163.651); t (4) = 0.125, *p* = 0.906;-Acetic acid solution (M = 410.800, SD = 136.844); t (4) = 1.304, *p* = 0.262;-Sulfuric + nitric acid solution (M = 409.000, SD = 140.260); t (4) = 1.257, *p* = 0.277.

In the charts from Figure 8 and Figure 9, the mean value Ra and Rq values are presented according to the solvent types.

The obtained results clearly show that there is not a significant difference between the roughness of the control group surface (untreated) and the roughness of each sample treated with different surfaces, and thus, the (H_0_) hypothesis is true. Therefore, it can be concluded that the composite synthetic plates are suitable for building ornaments from a chemical attack point of view.

We expected that our new material could resist acidic conditions. No significant statistical difference on pH decreasing proves that our composite is stable under harsh environmental conditions. The experimental results prove that the used polymer assures proper insulation of the calcite grains, preventing acid erosion. Therefore, the surface roughness is less affected by the pH increasing. It is one of the greatest successes of the current research.

The surface layer has a very beautiful ornamental aspect, being similar to marble due to the high calcite content of the exterior area. The polymer/calcite composite structure enhances wearing and weathering resistance in outdoor conditions. Thus, calcite powder micro- and even nano-scale distribution into the polymer could be the key to the observed resistance under weathering conditions. Another enhancement of the designed material is the structural layer having the role of consolidation to the surface area. The sand–grinded glass fiber–polymer system assures a good cohesion and enough mechanical strength. The natural sand used has a particle distribution in a range of 0 to 300 µm. Usually, sands are crystalline, having mainly quartz and several trace minerals such as calcite, clays, and residual oxides [50]. It is important to investigate the crystalline structure of the interior porous area (structural layer).

A thin slice of material (thickness of around 500 µm) was observed by cross-polarized light optical microscopy, resulting in the images in Figure 10. The exterior area shows a very good dispersion of calcite powder into the polymer matrix for all samples. The calcite-rich clusters have yellow shades and the polymer-rich areas have dark shades. The calcite clusters formed into the polymer range from 1 to 5 µm in diameter. The uniform illumination of calcite-rich clusters proves that there are no single crystals and must be polycrystalline for the possibility to have nanograins inside of them.

The structural layer is presented in Figure 11. The sand particles have a rounded/elongated shape with sharp edges for all samples. Most crystalline particles have a green-gray color, which is typical for quartz.

Several green particles presented intense green light iridescence as small spots of submicron size. These spots signal the presence of submicron features on a quartz particle surface. Some yellow-brown particles with diameters of around 50 µm are observed; these are traces of calcite naturally occurring in the sand. The dark areas in Figure 4b appear due to amorphous material such as glass fiber particles and polymers crusts. The data obtained by CPOM are centralized in Table 4.

CPOM analysis was coupled with XRD investigation for a complete crystalline characterization of the samples. The XRD spectra obtained for the exterior area are presented in Figure 12. All obtained diffraction peaks belong to calcite, proving the good quality of the sample.

The peaks present lower intensities as compared to those of a solid rock and present a slightly increased full width at half maximum (FWHM) of the XRD peaks. Applying the Scherrer formula to the FWHM, it results that crystal grains are nano-structured, and the values are given in Table 4. These values were confirmed by AFM microscopy; Figure 7. The XRD spectra baseline is slightly curved at low angles and the noise is increased due to the 60% amount of polymer.

An interesting behavior was observed for the sample exposed to the nitric acid mixture: the XRD sample features very low intensities for calcite peaks, almost mistaken for the baseline noise. It is a clue that suggests the disappearance of calcite grains from the sample surface as a consequence of acid erosion. AFM images for the samples exposed to rainwater and acetic acid feature submicron polymer clusters, with an average diameter of 300 nm, which contains nano calcite grains. In the sample exposed to the nitric acid mixture, the nano calcite grains vanish, with only empty polymer clusters remaining. This fact proves that polymer protects against acidic erosion of the surface of the slab. Data in the literature [49] show that nitric and sulfuric acid induces an intra-crystalline erosion of calcite grains, thus increasing surface roughness after long exposure.

The presence of polymer protects against chemical erosion of the interior strata of the exterior area, thus maintaining the surface roughness at a fair value.

XRD samples obtained for the structural layer are presented in Figure 13. Well-developed peaks appear, corresponding to a strong crystal structure. The dominant mineral is quartz (over 99%) and some calcite traces appear (below 1%) due to natural occurrence in sand, a fact sustained by CPOM microscopy. The XRD patterns prove that no changes occur in the crystalline structure of the samples. Thus, the crystal phase of the interior of the sample is resistant to erosion. In combination with the polymer and glass fiber, it resists well in weathering conditions.

It can be concluded that the surface of the material is only slightly affected in the case of HNO_3_ + H_2_SO_4_ and CH_3_COOH, while rainwater has no influence.

From Table 5, it can be observed that acid erosion has an influence on calcite crystal grains, as in the case of marble, but the presence of polymeric material stops the erosion propagation in the material.

The Scherrer formula applied to the quartz peaks shows crystal grains of around 110 nm diameters, a fact sustained by the AFM images, where fine submicron features appear on the bigger quartz particles. These are correlated with the bright green spots in the CPOM images in Figure 11. Such fine fractions of quartz are natural [50]. Their presence could be useful for the cohesion of the interior area because of interlocking with the polymer enhancing the bonding between bigger particles.

## 4. Conclusions

This paper presents a study regarding the evaluation of novel ornamental cladding resistance, comprised of GFRP waste and a polyester matrix. The wet material is pressed into a mold, and in the end, an ornamental plate that copies natural stone was obtained.The results of mechanical compression tests of the proposed material indicate 78 MPa. This value is comparable to C70/85 concrete. The proposed material solves a very important problem for composite factories. They accumulate large amounts of waste that are difficult to manage. Integrating GFRP waste and obtaining new materials can be a solution. The application of new materials also solves an important environmental problem. This may eliminate the environmental impact of stone quarries.The results of the performed analyses on the material regarding behavior upon possible chemical attacks when exposed to the environment show a very good behavior.The initial structures of both the composite plate and the ones that were acid attacked (acetic and nitric acids, which can be found in the mixing of rainwater with environmental pollution) were analyzed.

In order to check the surface modifications, SEM and AFM image analyses were conducted for each area of the prepared samples and for the interface between both components.

The morphology of the exterior part of the samples features a very good dispersion of calcite grains at the nano level, having a diameter around 70 nm as observed by XRD and AFM microscopy. These are optimally embedded into the polymer mass, which leads to a compact material as observed by SEM and AFM. The acidity increasing tends to erode calcite grains, slightly increasing their diameter to about 80 nm. The polymer stack on the calcite grains border prevents the erosion from expanding. The protective action is sustained, as seen by statistical analysis of the surface roughness, which is less affected.

The morphology of interior part of the sample (structural layer) is strongly related to the quartz particles interlocked with the polymer-based mortar. The structural cohesion is facilitated by the submicron quartz grains, as evidenced by XRD and AFM. Their diameter size is situated in the range of 110–120 nm, facilitating the mortar bonding to the bigger quartz particles observed with SEM and CPOM microscopy. Thus, good sealing assures against any acid infiltration into the structural layer.

Using software, the surface roughness values Ra and Rq were determined for each image. An independent samples t-test was conducted to compare surface roughness in untreated and treated surfaces for both situations.Following the statistical analysis of the roughness for both Ra and Rq cases, the obtained results clearly showed that there is no significant difference between the roughness of the control group surface (untreated) and the roughness of each sample treated with different substances for the exterior surface for the level α = 0.05 (*p*-value > 0.05 was obtained for all tests). This was also found in the case of the interior surface, meaning that rainwater does not affect the structure of the plate even if it is acid rain, or the plates are used in high-pollution environments. Components such as glass, polymer, and quartz particles are neutral from a chemical point of view.CPOM microscopy showed that even if calcite at micro- and nano-structural levels is soluble in acid solutions (which can appear due to rain), the presence of the polymer within the building plates reduces the erosion risk.SEM images also demonstrated that in the created porosity with different acid solutions, there was no capillarity to permit water infiltration, only some individual surface pores.This paper highlights the resistance properties of these composite plates in exterior environmental conditions. Taking into consideration that the exterior and interior surfaces are not significantly affected by these acid attacks, it can be concluded that the presented building plates are suitable for exterior building walls from a chemical attack point of view and provide resistance against compressive stresses. Therefore, these plates can be manufactured and used for cladding buildings.

## Figures and Tables

**Figure 1 polymers-13-00448-f001:**
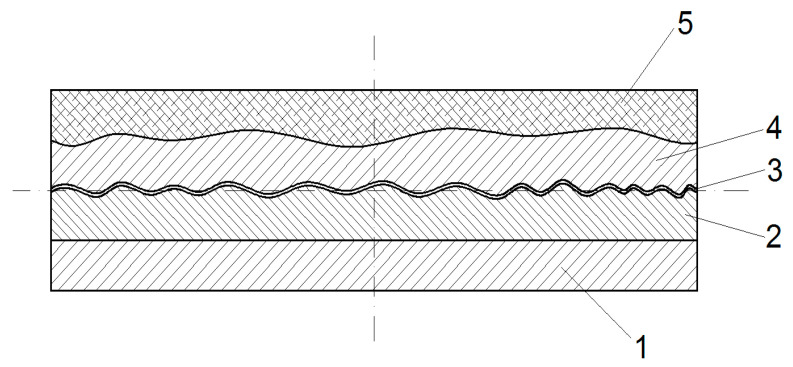
The main scheme to obtain ornamental synthetic plates [44]: 1—Mold support plate from glass fiber-reinforced polymer (GFRP); 2—Silicon rubber mold; 3—Release agent layer (polyvinyl alcohol, PVA); 4—Polyester resin with CaCO_3_; 5—Structural composite material: polyester resin, sand, and fiber glass waste.

**Figure 2 polymers-13-00448-f002:**
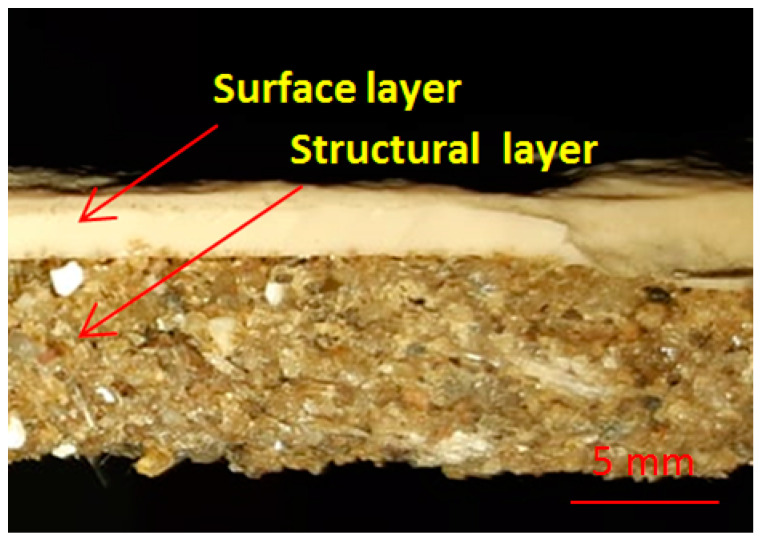
The ornamental synthetic stone plate: Interface between surface layer and structural composite material (CM).

**Figure 3 polymers-13-00448-f003:**
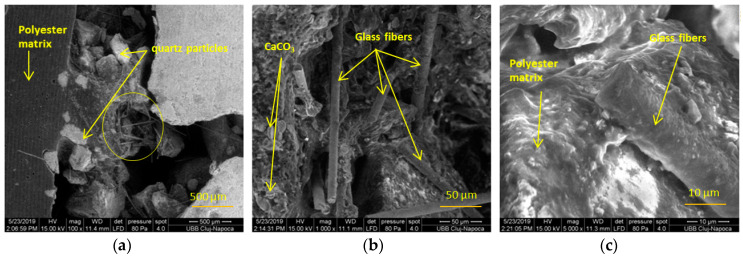
The SEM investigation of CM morphology surface (**a**–**c**).

**Figure 4 polymers-13-00448-f004:**
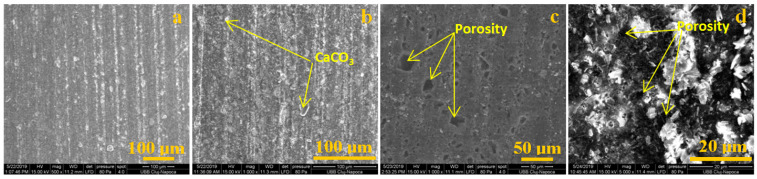
SEM images for exterior area: (**a**) control group; (**b**) exposed to rainwater; (**c**) exposed to acetic acid; (**d**) exposed to sulfuric and nitric acid.

**Figure 5 polymers-13-00448-f005:**
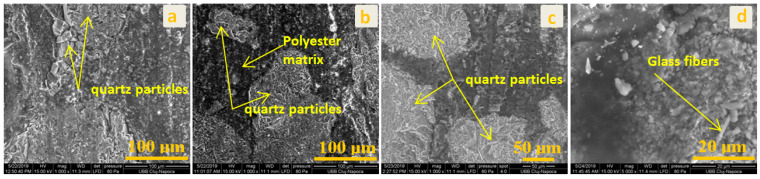
SEM images for structural material: (**a**) control group; (**b**) exposed to rainwater; (**c**) exposed to acetic acid; (**d**) exposed to sulfuric and nitric acid.

**Figure 6 polymers-13-00448-f006:**
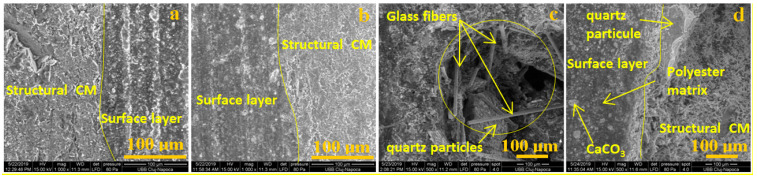
SEM images at the CM interface: (**a**) control group; (**b**) exposed to rainwater; (**c**) exposed to acetic acid; (**d**) exposed to sulfuric and nitric acid.

**Figure 7 polymers-13-00448-f007:**
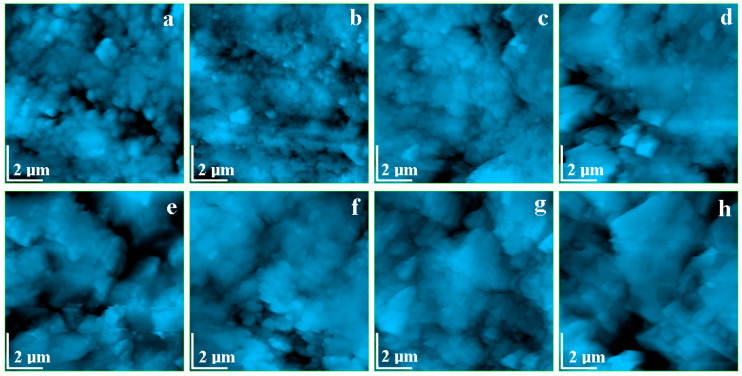
Atomic force microscopy (AFM) topographic images for surface layer: (**a**) control group; (**b**) exposed to rain water; (**c**) exposed to acetic acid; (**d**) exposed to sulfuric + nitric acid. For structural layer: (**e**) control group; (**f**) exposed to rain water; (**g**) exposed to acetic acid; (**h**) exposed to sulfuric + nitric acid. Scanned area 10 × 10 µm^2^.

**Figure 8 polymers-13-00448-f008:**
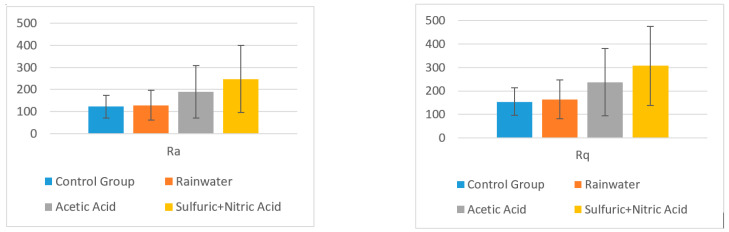
Mean values for Ra and Rq and their standard deviations in different solvents for surface layer.

**Figure 9 polymers-13-00448-f009:**
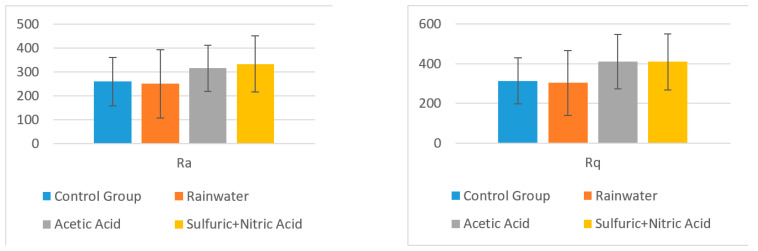
Mean values for Ra and Rq and their standard deviations in different solvents for structural layer.

**Figure 10 polymers-13-00448-f010:**
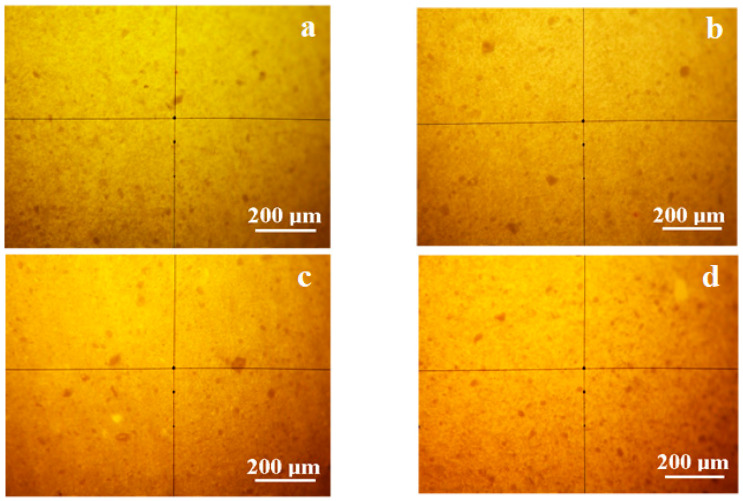
Cross-polarized light mineralogical microscopy images for surface layer: (**a**) control group; (**b**) rainwater; (**c**) acetic acid solution; (**d**) sulfuric + nitric acid solution.

**Figure 11 polymers-13-00448-f011:**
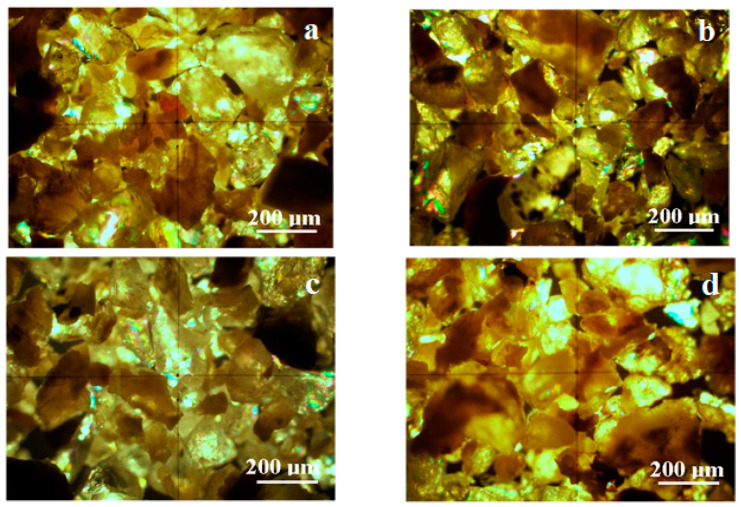
Cross-polarized light mineralogical microscopy images for structural layer: (**a**) control group; (**b**) rainwater; (**c**) acetic acid solution; (**d**) sulfuric + nitric acid solution.

**Figure 12 polymers-13-00448-f012:**
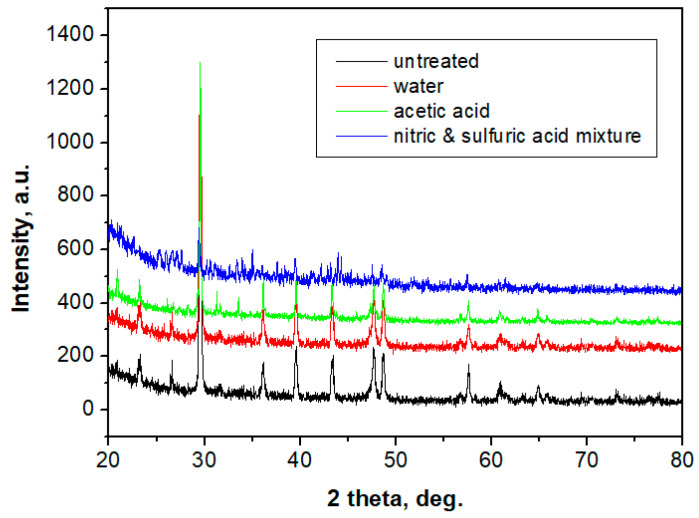
XRD patterns obtained for the exterior area of the: control group; group exposed to rain water; group exposed to acetic acid; and group exposed to nitric and sulfuric acid mixture. All XRD peaks belong to calcite.

**Figure 13 polymers-13-00448-f013:**
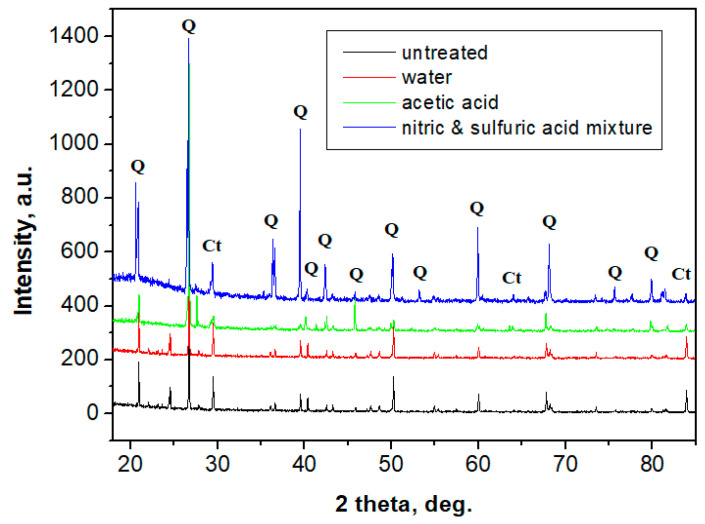
XRD patterns obtained for the interior porous area of the: control group; the group exposed to rainwater; group exposed to acetic acid; and group exposed to nitric and sulfuric acid mixture. (Q—quartz; Ct—calcite).

**Table 1 polymers-13-00448-t001:** Compressive tests results.

Sample Notation	Maximum Compressive Load (KN)	Average Compressive Load (KN)	Standard Deviation	Average Compressive Stress (MPa)
S1	192.6			
S2	188.8			
S3	192.4	190.76	1.76	78.89
S4	189.2			
S5	190.8			

**Table 2 polymers-13-00448-t002:** The roughness table for surface layer.

Measurement Points	Surface Layer			
	Roughness in 5 Different Points, Ra (μm)			
	Control Group	Group Test		
		Rainwater	Acetic Acid Solution	Sulfuric + Nitric Acid Solution
1	120.0	127.0	396.0	136.0
2	76.6	243.0	178.0	399.0
3	88.2	87.5	111.0	423.0
4	118.0	79.8	116.0	101.0
5	207.0	104.0	140.0	177.0
Mean	121.960	128.260	188.200	247.200
Std. Dev.	51.109	66.643	119.144	152.165
t-value		0.172762	1.186278093	1.82045214
*p*-value		0.871228	0.301158824	0.14280207
	**Roughness in 5 Different Points, Rq (μm)**			
1	164.0	166.0	484.0	191.0
2	95.2	304.0	233.0	470.0
3	113.0	109.0	143.0	505.0
4	152.0	101.0	147.0	137.0
5	246.0	142.0	179.0	232.0
Mean	154.04	164.40	237.20	307.00
Std. Dev.	58.54	82.28	142.59	168.64
t-value		0.236	1.252	2.015
*p*-value		0.8249	0.2786	0.1141

**Table 3 polymers-13-00448-t003:** The roughness table for structural layer.

Measurement Points	Structural Layer			
	Roughness in 5 Different Points, Ra (μm)			
	Control Group	Group Test		
		Rainwater	Acetic Acid Solution	Sulfuric + Nitric Acid Solution
1	309.0	264.0	463.0	298.0
2	193.0	455.0	363.0	297.0
3	164.0	165.0	275.0	244.0
4	219.0	293.0	246.0	286.0
5	411.0	73.9	231.0	540.0
Mean	259.200	250.180	315.600	333.000
Std. Dev.	100.728	143.445	96.984	117.792
t-value		0.118284	0.955398328	1.137632686
*p*-value		0.911544779	0.393479029	0.318793636
	**Roughness in 5 Different Points, Rq (μm)**			
1	367.0	318.0	625.0	370.0
2	244.0	525.0	466.0	369.0
3	203.0	199.0	355.0	310.0
4	262.0	373.0	299.0	340.0
5	492.0	98.3	309.0	656.0
Mean	313.600	302.660	410.800	409.000
Std. Dev.	116.637	163.651	136.844	140.260
t-value		0.125	1.304	1.257
*p*-value		0.90643	0.262095888	0.276984367

**Table 4 polymers-13-00448-t004:** Properties of minerals from exterior area and structural layer.

Mineral	Calcite	Quartz
Formula	CaCO_3_	SiO_2_
Cluster size, µm	1–5	10–400
Color	Yellow	Green-gray
Shape	Round, clusters	Round, elongated with sharp edges

**Table 5 polymers-13-00448-t005:** Crystal grain average diameter determined by XRD (Scherrer) and AFM.

	Crystal Grain Size in the Exterior Area, nm		Crystal Grain Size in the Interior—Porous Area, µm	
	XRD	AFM	XRD	AFM
Control group	68.44	73	110.68	120
Rainwater	69.84	75	110.49	120
Acetic acid	71.36	80	108.78	110
Azotic acid	-	-	109.46	115

## Data Availability

Not applicable.

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
