# Peer review of "Evaluation of Novel Ornamental Cladding Resistance, Comprised of GFRP Waste and Polyester Binder, within an Acid Environment"

_polymers, 2021, doi:10.3390/polym13030448_

Round 1

Reviewer 1 Report

Manuscript can be accepted after editing English.  

Reviewer 2 Report

Dear authors,

The manuscript presents some new and interesting results. However, several modifications are necessary:

  1. The title is not adequate since it does not inform the composite composition (polyester matrix and GFRP waste). This is also valid for the abstract.
  2. What was the composition of the GFRP waste (GF content)?
  3. Specify the percentage in each composition (mass percent or volume percent).
  4. What are the average thicknesses of the layers (surface and structural)?
  5. The porosities of each layer were quite high (up to 31%). Compare these values with previous similar studies.
  6. Add the errors (standard deviations) in table 1.
  7. Figures 3 to 6 should have the same magnification and scales, for comparison.
  8. Is Figure 7 (AFM) representative? The area investigated is so small (10 x 10 micrometers) that it could not represent the overall morphology.
  9. Although no statistical difference was observed, what the authors were expecting? Shouldn't the more acid solutions promote a higher roughness? What previous studies observed?
  10. Please write in full what means FWHM.
  11. In the conclusions, highlight the main implications of the study according to the results found.

Reviewer 3 Report

Application of waste for production of new and innovative building materials is one of the recent tendencies in civil engineering. Paper is interesting in this context.

There are only few recommendations for improvement of the paper.

1) Check spelling. There are few mistakes in the paper. Example at page 2: Write ' Thomas et al. [16] made a study ...' and not 'Thomas C. et al ...'

2) It os obvious that one sentence is one paragraph in Chapter 1. That is bad style. Please, check it.

3) It is proposed to apply the new developed composite building plates for construction of external walls and, may be, facade elements. One of the main attacks is sunlight and UV radiation. as authors write by themselves in Chapter 2.4. Otherwise, the authors only report experiments with acid attack. It not clear how these experiments cover UV radiation attack. Statement by the authors is needed in this context.

Reviewer 4 Report

This article deals with an interesting topic. However, Minor revision is required so that the article is easy to read and understand.

In general:

- Not all acronyms were introduced. Please, introduce the meaning of each acronym the first time it is proposed: for example SEM, AFM …

- Introduction: - It is suggested that the following articles be added to the cited articles:

https://doi.org/10.1016/j.conbuildmat.2015.12.173

https://doi.org/10.4028/www.scientific.net/KEM.624.421

- Section REFERENCES: Please, check the references. These have to be written in accordance with the journal instructions.

Round 2

Reviewer 2 Report

Dear editor,

The authors adequately tackled the points previously indicated, and the document was improved.

Although I believe that the document can be published, its scientific impact is low.

Best regards.